# MirrorCAPTCHA: Wild CAPTCHA, Wild Distribution, Wild Web-based Platform Meet Multimodal LLM Agents

## Abstract

The path to fully autonomous web agents is currently hindered by a critical bottleneck: their limited ability to handle CAPTCHA. Existing agent benchmarks largely ignore this practical challenge, failing to assess an agent's true capacity in cracking CAPTCHA. To bridge this gap, we comprehensively analyze the CAPTCHA distributions in the real world, and introduce **MirrorCAPTCHA** benchmark, annotated with *Weighted Pass Rate* and a novel proposed metric: *Completion Degree*. This benchmark is designed to serve as a "mirror" that accurately reflects the automation capabilities of agents in real scenarios. We filter out $2,095$ websites from the Common Crawl, identifying the active CAPTCHA puzzles and classifying them into $18$ distinct categories using the `K-means` clustering algorithm. To ensure practicality, we extract a web subgraph from Common Crawl covering these websites and employ random walks to simulate real-world CAPTCHA encounter frequencies, yielding a realistic measure of agents' ability. Additionally, we develop a lightweight synthetic data pipeline to train a model, `Ovis2-Agent-CAPTCHA-8B`, which significantly outperforms current state-of-the-art closed-source models on the MirrorCAPTCHA benchmark, achieving a $9.4\%$ higher average *Weighted Pass Rate* and a $2.13\%$ higher average *Completion Degree* compared with the second-place, `Gemini-2.5-Pro`.

## 1 Introduction

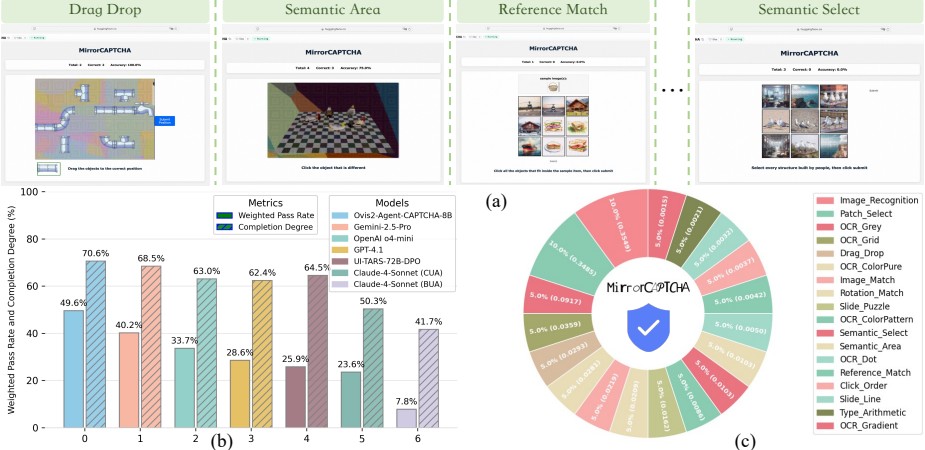

Figure 1: (a) Web-based CAPTCHA platform for evaluating web agents. (b) Performance of web agents. (c) MirrorCAPTCHA distribution. `CUA` and `BUA` denote Computer-Use Agents and Browser-Use Agents.

Multimodal web agents (He et al., 2024; Lai et al., 2024; Agashe et al., 2025; Huq et al., 2025), powered by multimodal large language models (MLLMs) (Wang et al., 2024; Lu et al., 2024; Chen et al., 2024), are designed to perform repetitive online tasks (*e.g.*, shopping, navigation, and booking), by simulating human behavior. However, a significant obstacle to their full automation is the requirement of CAPTCHA verification during common activities like registration and login. While agents can easily handle non-visual CAPTCHA (*e.g.*, SMS, email), the autonomous resolution of complex

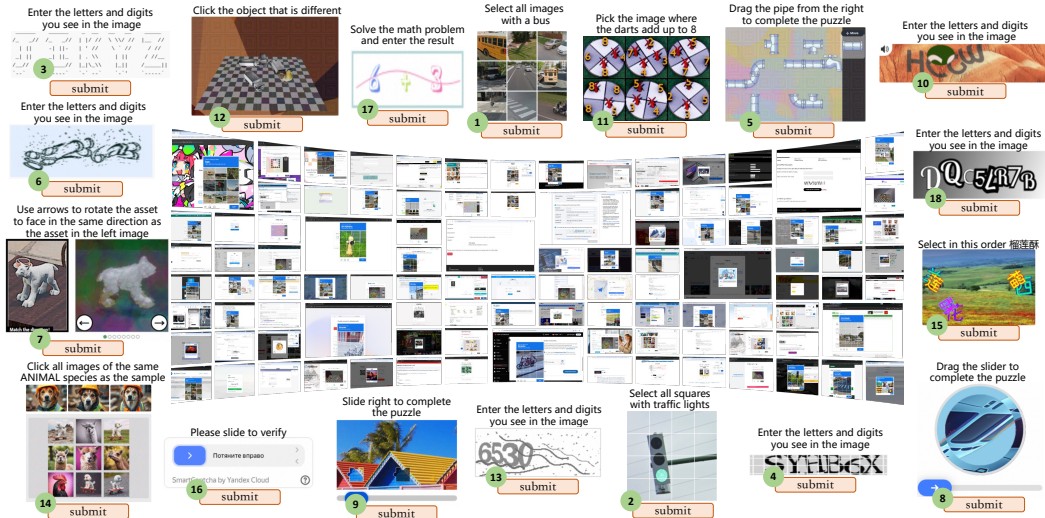

Figure 2: MirrorCAPTCHA filters 2095 valid websites with deployed CAPTCHAs from Common Crawl, covering 18 categories and 1000 puzzle samples, reflecting real-world CAPTCHA distribution.

visual challenges, such as grid selection, character recognition, and slider puzzles, remains an essential capability for their widespread deployment. Crucially, it remains unclear whether current agents can crack complex CAPTCHA in the wild with human-level speed and accuracy.

Mainstream web agent benchmarks (*e.g.*, VisualWebArena (Koh et al., 2024), AgentBench (Liu et al., 2024), and ST-WebAgentBench (Levy et al., 2025)) simulate real online environments but often omit prevalent CAPTCHA challenges. While recent works have introduced CAPTCHA-specific benchmarks, notable limitations persist. For instance, Open CaptchaWorld (Luo et al., 2025) introduces the first interaction-based benchmark but is limited by an extremely small dataset size, which fails to reflect real-world distributions and omits common Optical Character Recognition (OCR) puzzles. MCA-Bench (Wu et al., 2025) constructs a larger-scale, synthetic, homogeneous dataset that lacks practical realism, and some CAPTCHA types lack sufficient complexity, making accurate assessment of agent capabilities in the wild difficult.

To address these issues, we develop **MirrorCAPTCHA**, a benchmark designed to be a "mirror" of real-world CAPTCHA distribution and to accurately assess web agents' practical automation abilities. We filter out 2095 valid websites with active CAPTCHA puzzles from Common Crawl (Crawl, 2007), then classify them into 18 distinct categories, which comprise 1000 unique puzzles that span various deep learning and web interaction tasks, as shown in Figure 2. Notably, different CAPTCHA types are characterized by distinct frequency distributions, which are statistically derived through random walks on the web subgraph extracted from Common Crawl until a stable state is achieved. This ensures higher-frequency CAPTCHA types are assigned a larger weight in the evaluation. Consequently, the strong performance of an agent on MirrorCAPTCHA indicates its potential effectiveness in real-world scenarios.

Beyond the standard *Weighted Pass Rate* metric, we introduce a customized *Completion Degree* metric for part of the CAPTCHA types. While the pass rate measures binary success, the *Completion Degree* quantifies the "degree" to which an agent "solves" a CAPTCHA, offering a more nuanced measure of its reliability. All puzzles are tested on interactive webpages, as shown in Figure 1, to fully simulate the real-world scenarios agents encounter. Agents must perceive screenshots and perform actions like clicking, pressing keys, and dragging elements until the task is complete.

Additionally, we develop a lightweight and scalable data synthetic pipeline to train a model, `Ovis2-Agent-CAPTCHA-8B`. This model is trained on 370k synthetic CAPTCHA samples. Experiments on MirrorCAPTCHA show that `Ovis2-Agent-CAPTCHA-8B` significantly outperforms state-of-the-art closed-source models. For instance, on the high-traffic "Patch Select" category, our model surpasses `Gemini-2.5-Pro` by 30.66% in *Weighted Pass Rate*. The model's strong performance on both new metrics, including a high score in *Completion Degree* on challenging puzzles, highlights its potential for real-world web automation and sets a new state-of-the-art for multimodal agents on CAPTCHA challenges.

## 2 RELATED WORKS

**Web Agents**. Web Agents (Gur et al., 2024; He et al., 2024; Lai et al., 2024; Agashe et al., 2025; Huq et al., 2025; Shao et al., 2025; Erdogan et al., 2025), built upon large foundation models (Dubey et al., 2024; Yang et al., 2025), are designed to simulate human behavior and automate repetitive web tasks. These agents typically follow a three-step pipeline: perception (interpreting visual information from screenshots and text), planning/reasoning (decomposing tasks and generating actions), and execution (localizing elements and performing interactions). Recent advances, such as Auto-GPT (Significant Gravitas, 2023), demonstrate the ability to handle complex tasks with minimal user interaction. Similarly, multimodal agents like WebVoyager (He et al., 2024) and MMAC-Copilot (Song et al., 2024) leverage advanced models like GPT-4V (Yang et al., 2023) and Gemini Vision (Anil et al., 2023) to process diverse inputs, including screenshots and video content. Training strategies for these agents encompass data preprocessing, augmentation, and various fine-tuning methods, all of which aim to improve their end-to-end performance.

**Captcha Benchmarks and Models**. The development of deep learning has significantly advanced CAPTCHA recognition. Early methods relied on convolutional neural networks (CNNs) for feature extraction (Thobhani et al., 2020; Tang, 2024), while subsequent work combined CNNs and recurrent neural networks (RNNs) to handle variable-length CAPTCHA sequences (Hu et al., 2018; Derea et al., 2023). Generative adversarial networks (GANs) had also been used to synthesize large datasets for training CAPTCHA-cracking models (Shu & Xu, 2019; Ye et al., 2020). However, these models are often style-specific and lack the generalization required for real-world CAPTCHA variants. Existing benchmarks suffer from similar limitations. BeCAPTCHA-Mouse, for example, focuses on mouse trajectories with synthetic types, while Open CaptchaWorld (Luo et al., 2025) omits common Optical Character Recognition (OCR) CAPTCHA and has a small data size. MCA-Bench (Wu et al., 2025) evaluates vision-language models against synthetic, homogeneous CAPTCHA puzzles that do not reflect the diversity and complexity of real-world challenges. This gap, caused by a lack of benchmarks grounded in real-world distributions, prevents an accurate assessment of web agents' practical CAPTCHA-solving performance.

## 3 MIRRORCAPTCHA

MirrorCAPTCHA is a carefully curated benchmark of real-world CAPTCHA puzzles that are challenging for agents but easily solvable for humans. Most of the puzzles are directly collected from real websites and manually annotated, with a small portion sourced from MCA-Bench. The benchmark is evaluated using two metrics: *Weighted Pass Rate* (WPR) and a newly introduced metric, *Completion Degree* (CD), which applies to a fraction of CAPTCHA types.

### 3.1 DESKTOP WEB CURATION

MirrorCAPTCHA focuses on web agents that browse on desktop computers. To this end, the first step is to collect a large list of commonly visited, accessible desktop websites. Common Crawl (Crawl, 2007) provides a web graph of global internet traffic spanning the past six months, comprising $156.1$ million nodes and $2.1$ billion edges. Each node denotes a website accessed from a specific device (*e.g.*, desktop computer, phone), and each edge corresponds to a browsing transition.

From this graph, we select the top $15,000$ nodes based on degree as initial candidate sites. We then use a modified version `WebVoyager` (He et al., 2024) to query `Claude-4-Sonnet` for assessing their accessibility, filtering out inaccessible webpages (see Figure 3, top). The resulting corpus comprises $10,000$ valid websites spanning diverse domains, including entertainment, media, and social network platforms.

### 3.2 CAPTCHA-CONFRONTED WEB CURATION

The next step is to identify websites that trigger CAPTCHA mechanisms. Standard user actions (*e.g.*, direct registration or login by users) often do not trigger CAPTCHA, as such actions are typically not flagged as suspicious. Therefore, we deploy autonomous agents to systematically navigate and interact with registration and authentication workflows. This approach both increases the likelihood of triggering CAPTCHA challenges and reflects real-world challenges faced by web agents.

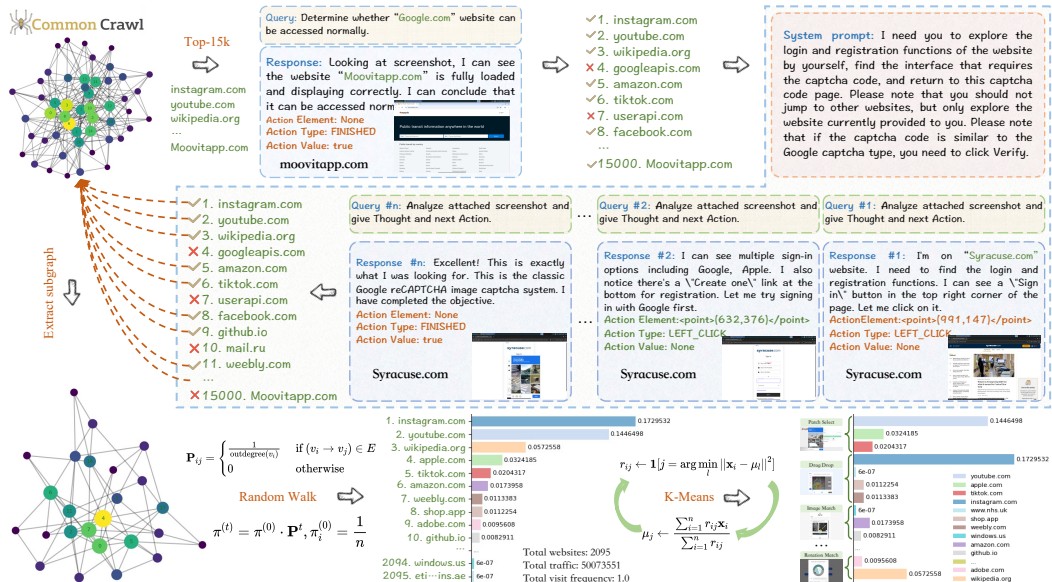

Figure 3: MirrirCAPTCHA construction pipeline. Top: modified WebVoyager querying `Claude-4-Sonnet` about website accessibility. Middle: `Claude-4-Sonnet` autonomously explores website functionalities that may trigger CAPTCHA (*e.g.*, registration, login, password reset, account recovery). Termination occurs when a CAPTCHA is triggered, the step limit is reached, or no CAPTCHA is confirmed. Bottom left: random walk for access probability estimation. Bottom right: K-means clustering for CAPTCHA categorization.

Figure 3 (middle) illustrates the entire free exploration process. We observe that if the agent fails to trigger a CAPTCHA on the current website, it will randomly navigate to other sites to continue searching for CAPTCHA challenges. Agents tend to force the completion of tasks regardless of the rationality of their execution. To mitigate this, we impose behavioral constraints:

- *Restricting exploration to registration/login interfaces*
- *Prohibiting site navigation beyond the given website*
- *Prioritizing interactions with web elements that could trigger CAPTCHA*

These rules ensure reliable activation and prevent redundancy. From the $10,000$ valid websites, we identify $2,095$ websites with CAPTCHAs deployed spanning multiple languages (English, Chinese, Russian), and diverse puzzle types (image/patch recognition, OCR, slides, drag-and-drop, arithmetic, and semantic tasks). While intuitive for humans, these puzzles remain difficult for agents.

Another critical consideration is that the real-world encounter frequency of a specific CAPTCHA type is directly determined by the traffic of websites that host it. For example, high-traffic platforms (e.g., Google, Facebook, YouTube), moderately popular niche sites with relatively lower traffic (e.g., GitHub, Adobe), and numerous obscure small websites with minimal traffic. Therefore, to accurately model the probability of a web agent encountering a particular CAPTCHA type, the benchmark must account for the traffic variations among the websites that deploy it.

## 3.3 WEB AND CAPTCHA ACCESS PROBABILITIES

To simulate real jumping behavior among websites, we perform random walks on the subgraph extracted from Common Crawl. Extracting nodes and edges adjacent to the $2,095$ websites yields a connected subgraph $\mathcal{G}_s$ with 15 million nodes and 75 million edges. Let $\mathbb{V} := \{v_1, v_2, ..., v_n\}$ denote its node set and $\mathbb{E} := \{e_1, e_2, ..., e_m\}$ denote its edge set. We define the transition matrix:

$$\mathbf{P}_{ij} = \begin{cases} \frac{1}{\text{outdegree}(v_i)} & \text{if } (v_i \rightarrow v_j) \in \mathbb{E}, \\ 0 & \text{otherwise,} \end{cases} \quad (1)$$

where outdegree($v_i$) denotes the number of outgoing edges from the node $v_i$. We initialize the visit probability distribution uniformly over the entire node set $\mathbb{V}$:

$$\pi_i^{(0)} = \frac{1}{n}, \quad \forall i = 1, 2, \ldots, n. \quad (2)$$

After $t$ steps, the node visit distribution is given by $\pi^{(t)} = \pi^{(t-1)} \cdot \mathbf{P}$. Recursively, we obtain $\pi^{(t)} = \pi^{(0)} \cdot \mathbf{P}^t$, with $\pi_i^{(t)}$ representing the probability of being at node $v_i$ after $t$ steps. After $10^8$ steps (the cutoff used in our study), the final visit probability distribution is:

$$\pi^{(10^8)} = \pi^{(0)} \cdot \mathbf{P}^{10^8}, \tag{3}$$

where $\pi_i^{(10^8)}$ is the probability of visiting node $v_i$. The bottom-left panel of Figure 3 illustrates the resulting traffic distribution, where a handful of high-traffic websites (*e.g.*, `instagram.com`, `youtube.com`, `wikipedia.org`) account for nearly half of all visits, and thus half of CAPTCHA encounters. *Detailed visit probability distribution for $\mathbb{V}$ is provided in Appendix C.*

Next, we categorize CAPTCHA types into clusters by applying `K-means` clustering. For each website screenshot $\mathrm{w}_i$, we extract the CLIP (Radford et al., 2021) image embedding $\mathbf{f}_i \in \mathbb{R}^d$:

$$\mathbf{f}_i = \phi(\mathrm{w}_i), \quad d = 512, \tag{4}$$

and stack the vectors to form:

$$\mathbf{F} = [\mathbf{f}_1, \mathbf{f}_2, \ldots, \mathbf{f}_N]^\top \in \mathbb{R}^{N \times d}, \quad N = 2095. \tag{5}$$

We then apply `K-means` to partition $\mathbf{F}$ into $K$ clusters $\{\mathrm{C}_1, \mathrm{C}_2, \ldots, \mathrm{C}_K\}$ via the standard iterative assignment-and-update procedure:

$$r_{ij} \leftarrow \mathbf{1}\left[ j = \arg\min_l \|\mathbf{f}_i - \boldsymbol{\mu}_l\|^2 \right],$$

$$\boldsymbol{\mu}_j \Leftarrow \frac{\sum_{i=1}^N r_{ij}\mathbf{f}_i}{\sum_{i=1}^N r_{ij}}, \tag{6}$$

where $\boldsymbol{\mu}_j$ denotes the centroid of cluster $\mathrm{C}_j$, and $r_{ij}$ is an indicator of whether $\mathrm{w}_i$ belongs to $\mathrm{C}_j$. Starting from $K = 2$, we iteratively refine the clustering, manually examine the clustering results, and further partition more fine-grained clusters. Ultimately, we obtain 18 distinct CAPTCHA types, as shown in the bottom-right panel in Figure 3.

Once the clustering is completed, we compute the visit probability of each CAPTCHA category $\mathrm{C}_j$ by aggregating the visit probability $\pi_i^{(10^8)}$ of all websites in that cluster:

$$p(\mathrm{C}_j) = \sum_{\mathrm{w}_i \in \mathrm{C}_j} \pi_i^{(10^8)}, \quad j = 1, 2, \ldots, K. \tag{7}$$

The resulting distribution of access frequencies across the 18 CAPTCHA categories is summarized in Table 1, where the categories show a heavy-tailed pattern $-$ a few dominant types (such as distorted alphanumeric text or simple image-based challenges) account for the majority of real-world traffic, whereas many others are far less prevalent. This skewed distribution will directly impact the design of evaluation datasets and robustness benchmarks for automated CAPTCHA solvers.

The final step is to construct puzzle samples for each CAPTCHA type in proportion to visit frequency: categories with higher traffic are allocated more samples, thereby mirroring real-world CAPTCHA distribution patterns. However, some categories (*e.g.*, OCR Gradient, Type Arithmetic) exhibit extremely low traffic. For instance, in a benchmark with $1,000$ samples, OCR Gradient would yield only 1-2 puzzles ($1,000 \times 0.00148$), which is overly sparse. Conversely, Image Recognition or Patch Selection may dominate with hundreds of samples, leading to redundancy.

To balance realism and robustness, we cap the number of samples per category at 50 or 100, as shown in Table 1. *See Appendix A for the comparison with OpenCaptchaWorld and MCA-Bench.* During evaluation, the true visit frequencies remain as weights when aggregating results, ensuring that the *Weighted Pass Rate* reflects real-world CAPTCHA distribution while avoiding extreme sparsity or overrepresentation. Details of this measure strategy are discussed in the following subsection.

### 3.4 EVALUATION METRICS

MirrorCAPTCHA employs two metrics: *Weighted Pass Rate* (WPR) and *Completion Degree* (CD). WPR measures whether a model fully solves a CAPTCHA puzzle, weighted by real-world encounter probabilities, and CD quantifies how close a model comes to a full solution. All CAPTCHA types can be assessed with WPR, whereas only a subset is compatible with it. For example, Image Match puzzles are strictly binary (match or non-match) and therefore can only be evaluated using WPR.

Table 1: Statistics of the MirrorCAPTCHA benchmark by category, including website coverage, visit traffic, relative frequency, example puzzles, task description, and number of samples. Categories are ordered by traffic: 1. Image Recognition, 2. Patch Select, 3. OCR Grey, 4. OCR Grid, 5. Drag Drop, 6. OCR ColorPure, 7. Image Match, 8. Rotation Match, 9. Slide Puzzle, 10. OCR ColorPattern, 11. Semantic Select, 12. Semantic Area, 13. OCR Dot, 14. Reference Match, 15. Click Order, 16. Slide Line, 17. Type Arithmetic, 18. OCR Gradient.

| Type name | Covered | Traffic | Frequency | CAPTCHA description | Samples |
|---|---|---|---|---|---|
| Image Recognition | 652 | 17961686 | 0.35871 | Identify target objects grid in a 9-image grid | 100 |
| Patch Select | 709 | 17449069 | 0.34847 | Identify target objects patches in a 16-image grid | 100 |
| OCR Grey | 95 | 4594340 | 0.09175 | OCR: grayscale text, and line noise | 50 |
| OCR Grid | 22 | 1798535 | 0.03592 | OCR: grayscale text, grid background, and line noise | 50 |
| Drag Drop | 58 | 1465736 | 0.02927 | Drag small image to correct position on large image | 50 |
| OCR ColorPure | 159 | 1408947 | 0.02814 | OCR: color font, pure background, and color line noise | 50 |
| Image Match | 18 | 1094478 | 0.02186 | Select matching image from candidates based on reference | 50 |
| Rotation Match | 6 | 1047564 | 0.02092 | Rotate tile to correct position via slider | 50 |
| Slide Puzzle | 103 | 811840 | 0.01621 | Slide puzzle piece to correct position | 50 |
| OCR ColorPattern | 46 | 428499 | 0.00856 | OCR: color font, pattern background, and color line noise | 50 |
| Semantic Select | 50 | 516790 | 0.01032 | Select images from 3×3 grid following instructions | 50 |
| Semantic Area | 34 | 514516 | 0.01027 | Select the different icon from multiple similar ones | 50 |
| OCR Dot | 57 | 248757 | 0.00497 | OCR: grayscale text, pockmarked background, and line noise | 50 |
| Reference Match | 26 | 209916 | 0.00419 | Select from 3×3 grid based on references and instructions | 50 |
| Click Order | 15 | 182529 | 0.00365 | Click icons in specified sequence | 50 |
| Slide Line | 13 | 159555 | 0.00319 | Slide block to endpoint | 50 |
| Type Arithmetic | 15 | 106586 | 0.00213 | Solve arithmetic problem and enter result | 50 |
| OCR Gradient | 17 | 74208 | 0.00148 | OCR: grayscale font, gradient background, and line noise | 50 |
| **Total** | 2095 | 50073551 | 1.0 | – | 1000 |

**Weighted Pass Rate (WPR).** The visit probability $p(C_j)$ of a given CAPTCHA category is defined in Equation 7. Let $N_i$ denote the total number of puzzle samples in category $i$, and $S_i$ denote the number of puzzles that the model fully and correctly solves. Then:

$$\textbf{WPR} = \sum_{i=1}^{k} \left( p_i \times \frac{S_i}{N_i} \right) \times 100\% \tag{8}$$

**Completion Degree (CD).** CD is defined for 12 categories using 4 task-specific measures (*See Appendix B for detailed evaluation metrics*):

- `F1 score` (van Rijsbergen, 1979): Applied to Image Recognition, Patch Select, Semantic Select, and Reference Match; computes the F1 score between model predictions and ground truth labels.

- `Levenshtein Distance` (Levenshtein, 1966): Applied to OCR Grey, OCR Gradient, OCR Grid, OCR ColorPure, OCR ColorPattern, and OCR Dot; measures edit distance between predicted and true strings.

- `Sequence Matching`: Applied to Click Order; counts one-to-one matches between the predicted and ground truth sequences.

- `Angle Distance`: Applied to Rotation Match; measures angular difference between predicted and ground truth orientations.

Taken together, WPR and CD offer a holistic evaluation of CAPTCHA-solving performance, capturing both strict accuracy and partial progress.

## 4 Ovis2-Agent-CAPTCHA-8B

Unlike prior deep learning models that target a single CAPTCHA type (*e.g.*, $3 \times 3$ or $4 \times 4$ grids) with task-specific designs, the MirrorCAPTCHA benchmark evaluates the broader capability of MLLM-based web agents to solve diverse, real-world CAPTCHAs. This naturally raises the question: *how can we enhance a web agent's ability to generalize across a wide range of CAPTCHA types?*

To address this, we design a lightweight, extensible CAPTCHA synthesis pipeline in Figure 4 that can automatically generate puzzles by defining certain rules for data organization. Leveraging this synthesized data, we apply supervised fine-tuning (SFT) to Ovis2-8B (Lu et al., 2024), yielding `Ovis2-Agent-CAPTCHA-8B`. `Ovis2-Agent-CAPTCHA-8B` is trained with SFT on 370k synthesized CAPTCHA samples (including Image Recognition, Patch Select, Semantic Area, OCR, and Type Arithmetic types), augmented with limited computer-use trajectory data to improve cross-scenario adaptability. Training requires 35 hours on 64 H100 GPUs, enabling the model to acquire the necessary skills in visual grounding, semantic reasoning, and interactive operation for CAPTCHA solving.

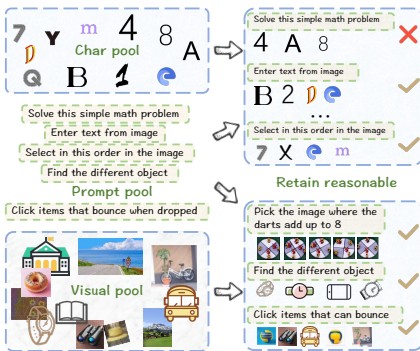

Figure 4: Synthesize CAPTCHA pipeline.

We then benchmark the model against both open-source and closed-source counterparts. Experimental results show that `Ovis2-Agent-CAPTCHA-8B` not only surpasses existing open-sourced models but also significantly outperforms state-of-the-art closed-sourced systems on MirrorCAPTCHA, setting a new technical baseline for CAPTCHA-solving agents.

# 5 EXPERIMENTS ANALYSIS

## 5.1 EXPERIMENTAL SETUP

We systematically evaluate both browser-use agents and computer-use agents, each equipped with different MLLM backbones, on the MirrorCAPTCHA benchmark. To ensure fairness, we adopt consistent prompting strategies and uniform evaluation metrics across all models. **Browser-use agents**, implemented with the set-of-mark (SOM) paradigm (Müller & Žunič, 2024), include OpenAI o4-mini (OpenAI, 2025), Gemini-2.5-Pro (Anil et al., 2023), Claude-4-Sonnet (Anthropic, 2025), and GPT-4.1 (OpenAI, 2025). **Computer-use agents** are deployed via the augmented Web-Voyager framework (He et al., 2024), covering Claude-4-Sonnet (Anthropic, 2025), UI-TARS-72B-DPO (Qin et al., 2025), and our Ovis2-Agent-CAPTCHA-8B model.

Table 2: WPR on MirrorCAPTCHA for Browser-Use (OpenAI o4-mini, Gemini-2.5-Pro, Claude-4-Sonnet, GPT-4.1) and Computer-Use Agents (Claude-4-Sonnet, UI-TARS-72B-DPO, Ovis2-Agent-CAPTCHA-8B).

| CAPTCHA Type | Browser-Use Agent | | | | Computer-Use Agent | | |
| --- | --- | --- | --- | --- | --- | --- | --- |
| | o4-mini | Gemini-2.5-Pro | Claude-4-Son | GPT-4.1 | Claude-4-Son | UI-TARS | Ovis2-8B |
| Image Recognition | 53.87 | 64.33 | 3.72 | 47.67 | 35.33 | 40.07 | **66.67** |
| Patch Select | 14.72 | 14.67 | 2.76 | 4.67 | 10.00 | 5.39 | **45.33** |
| OCR Grey | 50.00 | **62.00** | 30.00 | 54.00 | 38.00 | 48.00 | 54.00 |
| OCR Grid | 36.00 | **54.00** | 20.00 | 44.00 | 22.00 | 37.50 | 52.00 |
| Drag Drop | 0.00 | 0.00 | 0.00 | 0.00 | 0.00 | 0.00 | 0.00 |
| OCR ColorPure | 44.00 | **62.00** | 24.00 | 54.00 | 32.00 | 40.00 | 50.00 |
| Image Match | 6.67 | **31.03** | 14.29 | 13.33 | 30.00 | 23.33 | 6.67 |
| Rotation Match | 0.00 | 0.00 | 0.00 | 0.00 | 0.00 | **4.00** | 0.00 |
| Slide Puzzle | 0.00 | 0.00 | 0.00 | 0.00 | 0.00 | **4.00** | 0.00 |
| OCR ColorPattern | 62.00 | **74.00** | 28.00 | 64.00 | 32.00 | 52.00 | 60.00 |
| Semantic Select | 30.00 | **56.67** | 24.21 | 43.33 | 46.67 | 33.33 | 26.67 |
| Semantic Area | 0.00 | 0.00 | 0.00 | 0.00 | 40.00 | **53.33** | 16.67 |
| OCR Dot | 50.00 | 70.00 | 32.00 | 54.00 | 30.00 | 50.00 | **74.00** |
| Reference Match | 50.00 | **83.33** | 33.33 | 33.33 | 0.00 | 23.33 | 10.00 |
| Click Order | 0.00 | 0.00 | 0.00 | 0.00 | 0.00 | **3.75** | 1.25 |
| Slide Line | 0.00 | 0.00 | 0.00 | 0.00 | 50.00 | **80.00** | 20.00 |
| Type Arithmetic | 93.33 | **96.67** | 90.00 | 90.00 | 90.00 | 80.00 | **96.67** |
| OCR Gradient | 64.00 | **70.00** | 34.00 | 62.00 | 40.00 | 60.00 | 64.00 |
| Average | 33.66 | 40.22 | 7.77 | 28.58 | 23.58 | 25.85 | **49.57** |

## 5.2 WEIGHTED PASS RATE AND COMPLETION DEGREE

Table 2 reports the WPR for all models. Due to inherent limitations of the browser-use execution framework (Müller & Žunič, 2024), such agents cannot perform operations like "Drag Drop", "Rotation match", "Slide Puzzle", "Semantic Area", "Click Order" or "Slide Line". Specifically, the SOM mechanism treats an image as a single object and cannot localize or manipulate elements within it. By contrast, computer-use agents directly simulate mouse interactions, enabling them to attempt all CAPTCHA categories.

**Weighted Pass Rate.** Ovis2-Agent-CAPTCHA-8B achieves the highest average WPR. Notably, on the high-traffic "Patch Select" category, it surpasses Gemini by 30.66 percentage points. For Claude-4-Sonnet (computer-use), WPR drops to zero in some categories, largely due to sensitivity to image resolution, which prevents reliable coordinate output for click actions. For Claude-4-Sonnet (browser-use), failures in "Image Recognition" and "Patch Select" stem from repeated memory read/write loops, reflecting a fundamental vulnerability of the framework.

Table 3: CD on MirrorCAPTCHA for Browser-Use (OpenAI o4-mini, Gemini-2.5-Pro, Claude-4-Sonnet, GPT-4.1) and Computer-Use Agents (Claude-4-Sonnet, UI-TARS-72B-DPO, Ovis2-Agent-CAPTCHA-8B).

| CAPTCHA Type | Browser-Use Agent | | | | Computer-Use Agent | | |
|---|---|---|---|---|---|---|---|
| | o4-mini | Gemini-2.5-Pro | Claude-4-Son | GPT-4.1 | Claude-4-Son | UI-TARS | Ovis2-8B |
| Image Recognition | 84.27 | 80.88 | 22.83 | 66.77 | 76.22 | 72.11 | **89.66** |
| Patch Select | 72.82 | 70.22 | 33.49 | 57.53 | 61.60 | 57.24 | **88.14** |
| OCR Grey | 81.53 | **84.52** | 65.84 | 82.52 | 70.41 | 76.08 | 78.84 |
| OCR Grid | 76.13 | **84.69** | 56.21 | **78.56** | 60.26 | 72.68 | 78.10 |
| OCR ColorPure | 75.66 | **85.30** | 62.15 | 81.41 | 67.47 | 75.60 | 79.14 |
| Rotation Match | 0.00 | 0.00 | 0.00 | 0.00 | 0.00 | **60.97** | 60.27 |
| OCR ColorPattern | 85.20 | **89.83** | 64.52 | 86.20 | 68.26 | 76.15 | 79.53 |
| Semantic Select | 49.48 | 62.22 | 31.11 | **64.32** | 60.37 | 55.95 | 61.25 |
| OCR Dot | 76.00 | **89.23** | 60.54 | 82.85 | 67.78 | 77.49 | 88.33 |
| Reference Match | 69.68 | **89.44** | 39.32 | 58.97 | 0.00 | 55.95 | 49.27 |
| Click Order | 0.00 | 0.00 | 0.00 | 0.00 | 0.00 | **11.46** | 10.62 |
| OCR Gradient | 85.63 | 85.63 | 65.15 | **89.33** | 71.41 | 82.54 | 84.36 |
| Average | 63.03 | 68.50 | 41.68 | 62.37 | 50.32 | 64.52 | **70.63** |

**Completion Degree.** Table 3 shows that Ovis2-Agent-CAPTCHA-8B also achieves the best average CD. Interestingly, in categories such as "Patch Select", the WPR is relatively low across all models, while CD remains high, indicating that models select most patches correctly but often fail to complete the task perfectly. This suggests that higher image resolution, more query steps, or finer-grained behavioral optimization could close the gap between CD and WPR.

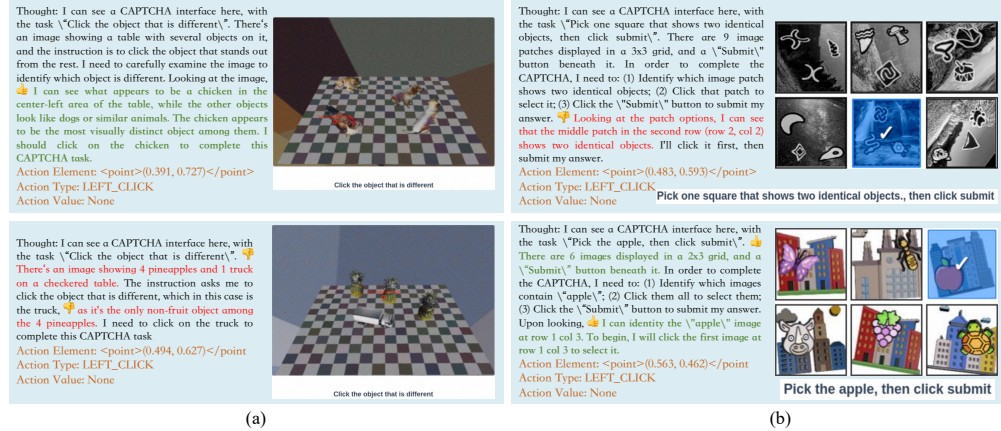

(a)  (b)

Figure 5: Correct vs. error cases for Ovis2-Agent-CAPTCHA-8B on (a) "Click the object that is different" and (b) "Semantic Select".

### 5.3 SUCCESS AND FAILURE CASE ANALYSIS

Figure 5 presents typical success and failure cases. In the correct case of the "Click the object that is different" task, the model identifies a chicken among dog-like animals and clicks the correct location. In the error case, however, it misclassifies the objects as 4 pineapples and 1 truck. Although it correctly identifies the truck as the outlier, the click misses the precise coordinates, causing task failure. Similarly, in the error case, failure arises from difficulty in interpreting abstract icons. Here, the model guesses randomly rather than reasoning about semantic differences.

These cases highlight two key limitations: **(i)** imprecise visual recognition of object attributes and counts, and **(ii)** difficulty in extracting discriminative features from abstract images. Addressing these issues will require improved visual feature extraction and more reliable object classification.

### 5.4 INFERENCE PROCESS ON CHAALENGING CAPTCHA PUZZLES

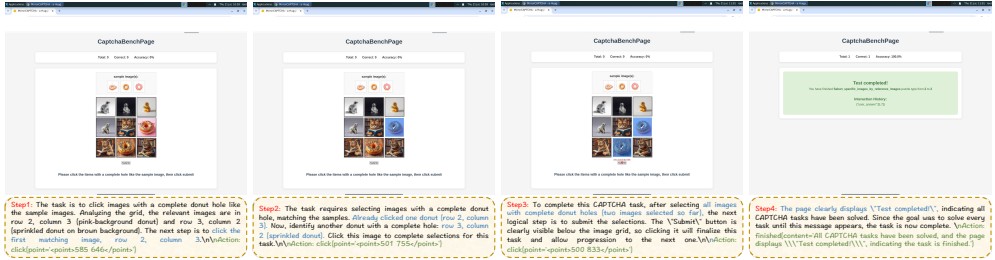

Figure 6: CAPTCHA puzzle: Click the items with a complete hole, as shown in the sample images.

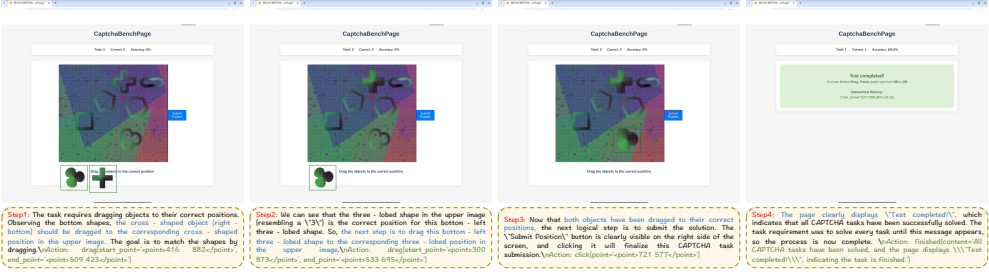

Figure 7: CAPTCHA puzzle: Drag the objects to their correct position.

Figures 6 and 7 provide a qualitative look into how web agent solves challenging CAPTCHA on the MirrorCAPTCHA benchmark. First, it accurately interprets the natural language instructions provided by the puzzle, such as identifying the required criteria ("items with a complete hole") or understanding the action needed ("Drag the objects to their correct position"). The agent then applies its visual reasoning skills to identify the correct target elements on the webpage. Finally, it translates this understanding into a series of precise interactive operations, including clicks, drags, and text entry, to complete the CAPTCHA task. This seamless integration of perception, reasoning, and execution highlights the agent's advanced capabilities. *More detailed examples of the agent's reasoning process are provided in Appendix D.*

## 6 CONCLUSION

We present **MirrorCAPTCHA**, a benchmark designed to act as a "mirror" of real-world CAPTCHA distributions. It filters $2,095$ valid websites with deployed CAPTCHAs from Common Crawl, categorized into 18 types spanning both deep learning and web interaction tasks. To approximate real-world encounter frequencies, MirrorCAPTCHA employs random walks and evaluates performance using two metrics: *Weighted Pass Rate* and *Completion Rate*. In addition, we introduce `Ovis2-Agent-CAPTCHA-8B`, a model trained on a synthesized CAPTCHA dataset. Experimental results show that it significantly outperforms both open-source and closed-source counterparts, surpassing Gemini-2.5-Pro by $9.4\%$ in Weighted Pass Rate and achieves the highest Completion Degree across most CAPTCHA categories.

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

# Appendix for MirrorCAPTCHA

## A COMPARISON WITH EXISTING CAPTCHA DATASETS

Table 4: Comparison with Open CaptchaWorld and MCA-Bench benchmarks.

| Dataset | CAPTCHA Distribution | Number of Categories | Data size | CAPTCHA Categories |
|---------|---------------------|----------------------|-----------|--------------------|
| Open CaptchaWorld | Random | 20 | 225 | Select Animal, Pick Area, Patch Select, Object Match, Misleading Click, Geometry Click, Image Recognition, Coordinates, Place Dot, Rotation Match, Image Matching, Connect Icon, Bingo, Dart Count, Dice Count, Slide Puzzle, Path Finder, Click Order, Unusual Detection, Hold Button |
| MCA-Bench | Random | 20 | 4,000 | $3 \times 3$ Grid Select, $3 \times 3$ Jig-swap, Arithmetic Char, Arithmetic Select, Hollow Pattern, Distort Word, Classic Char, Sequential Letter, Bright Dist, Sliding Block, Align Sliders, Rotate Block, Geometry Shape, Rotation Letter, Color Discrimination, Vowel Select, Full-img Grid Select, Text-based Arithmetic, Common Sense, Invert Letter |
| **MirrorCAPTCHA** | Real-world | 18 | 1,000 | Image recognition, Patch Select, OCR Grey, OCR Grid, Drag Drop, OCR Color-pure, Image Match, Rotation Match, Slide Puzzle, OCR Color-pattern, Semantic Select, Semantic Area, OCR Dot, Reference Match, CLICK Order, Slide Line, Type Arithmetic, OCR Gradient |

Table 4 presents a detailed comparison of the three most recent CAPTCHA benchmarks. Both Open CaptchaWorld (Luo et al., 2025) and MCA-Bench (Wu et al., 2025) rely on a random distribution of CAPTCHA types, which fails to accurately reflect the real-world frequencies of these challenges. Open CaptchaWorld includes 20 categories but has an extremely small dataset of just 225 samples, which can lead to high randomness in evaluation. MCA-Bench is larger, with 4,000 samples across 20 categories, but its synthetic puzzles do not capture the diversity and complexity found on real websites. In contrast, our MirrorCAPTCHA benchmark is grounded in real-world data. It features 18 CAPTCHA types collected from 2,095 live websites, and its distribution is statistically derived from actual web traffic. With a total of 1,000 puzzles, MirrorCAPTCHA provides a more realistic and reliable tool for evaluating web agents, making it a "mirror" that truly reflects an agent's performance in real-world scenarios.

## B DETAILED EVALUATION METRIC: COMPLETION DEGREE

To capture partial correctness in CAPTCHA-solving tasks, we define `Completion Degree` (CD) as a family of fine-grained metrics that quantify *how close* a web agent's output is to the correct answer, even when the CAPTCHA is not fully solved.

We adopt four types of CD metrics: `F1 Score`, `Levenshtein Distance`, `Sequence Matching Accuracy`, and `Angle Distance Error`, each aligned with the nature of specific CAPTCHA categories.

▶ `F1 Score`

This metric is used for puzzles that require selecting one or more items from a set, such as:

- Image Recognition
- Patch Select
- Semantic Select
- Reference Match

**Definition**. The `F1 score` is the harmonic mean of precision and recall:

$$F_1 = \frac{2 \times \text{Precision} \times \text{Recall}}{\text{Precision} + \text{Recall}},$$

where

$$\text{Precision} = \frac{\text{TP}}{\text{TP} + \text{FP}}, \quad \text{Recall} = \frac{\text{TP}}{\text{TP} + \text{FN}},$$

where TP (True Positive) denotes correctly selected items, FP (False Positive) denotes wrongly selected items, and FN (False Negative) denotes missed correct items.

**Explanation**: These CAPTCHA types may contain multiple correct elements. F1 score balances correctness and completeness: selecting wrong patches lowers precision, while missing correct ones lowers recall. F1 scores are normalized to $[0, 1]$.

▶ Levenshtein Distance

This metric is used for CAPTCHAs that involve text recognition:

- OCR Grey
- OCR Gradient
- OCR Grid
- OCR ColorPure
- OCR ColorPattern
- OCR Dot

**Definition**. The `Levenshtein distance` measures the minimum number of single-character edits (insertions, deletions, substitutions) needed to transform the predicted string $s_1$ into the ground truth $s_2$:

$$\text{Lev}(s_1, s_2) \in \mathbb{N}^+.$$

We convert this distance into a similarity score:

$$\text{CD} = 1 - \frac{\text{Lev}(s_1, s_2)}{\max(|s_1|, |s_2|)}.$$

A score of 1 indicates a perfect match, while a score of 0 indicates completely different strings.

**Explanation**. This metric captures OCR-specific errors (*e.g.*, character substitutions) and awards partial credit when most of the string is correct.

▶ Sequence Matching Accuracy

This metric is specifically designed for:

- Click Order

**Definition**. Let $P$ be the sequence of elements clicked by the agent and $G$ the ground truth sequence. We compute:

$$\text{CD} = \frac{1}{n} \sum_{i=1}^{n} \mathbf{1}\{P_i = G_i\},$$

where $n$ is the sequence length, $P_i$ is the $i$-th clicked element, and $\mathbf{1}\{\cdot\}$ is the indicator function.

**Explanation**. This metric measures position-wise accuracy. For example, if the CAPTCHA requires clicking "Cat → Dog → Bird", this metric counts how many items are in the correct position. This allows for partial credit even if the model only gets part of the sequence right.

▶ Angle Distance Error

This metric is used for puzzles that require rotation:

- Rotation Match

**Definition**. Given predicted rotation $\theta_p$ and ground truth $\theta_g$, we compute:

$$\Delta\theta = \min\left(|\theta_p - \theta_g|, 360 - |\theta_p - \theta_g|\right),$$

which correctly accounts for circular periodicity (*e.g.*, $0° \equiv 360°$). The completion score is defined as:

$$\text{CD} = 1 - \frac{\Delta\theta}{\theta_{\max}},$$

where $\theta_{\max}$ is the maximum possible deviation ($180°$). A CD of 1 means the rotation is perfectly aligned, 0.5 means a misalignment of $90°$, and 0 means an opposite alignment ($180°$ difference).

**Explanation**. This metric gives proportionate credit for predictions that are close to the correct angle, which more accurately reflects an agent's capability in near-solved cases compared to a binary pass/fail judgment.

## C  DETAILED VISIT PROBABILITY OF WEBSITES

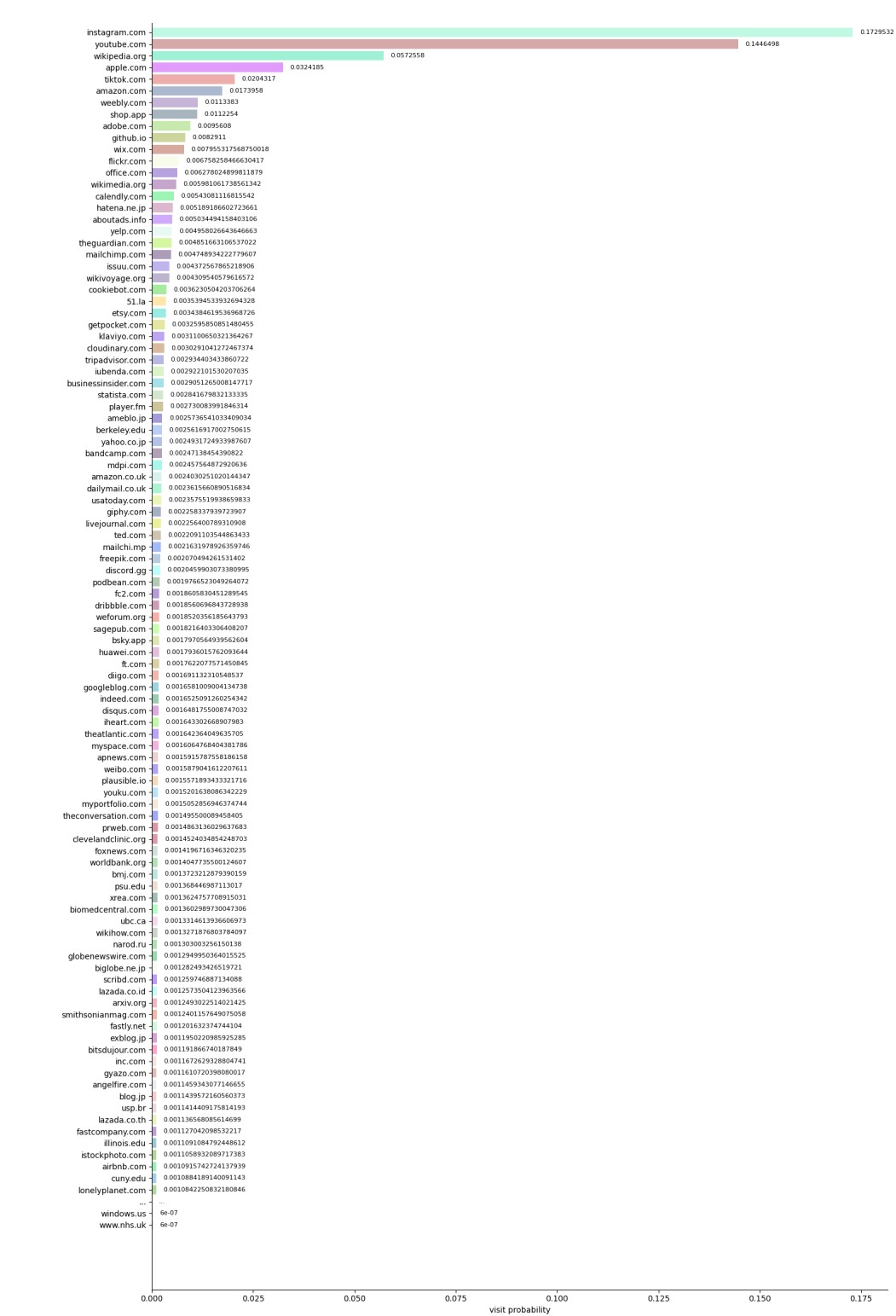

Figure 8: Detailed visit frequencies distribution of the 2, 095 websites with deployed CAPTCHAs. The top 10 websites account for nearly half of the total traffic, reflecting real-world internet patterns.

# D    ADDITIONAL REASONING PROCESS ON MIRRORCAPTCHA

We provide further examples of CAPTCHA-solving processes to illustrate how the web agent interacts with real-world webpages. These cases highlight the agent's ability to interpret and reason over CAPTCHA puzzles, make decisions, and execute corresponding interactive actions.

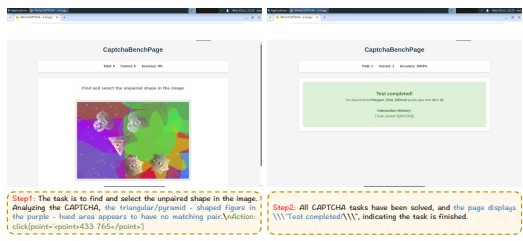

Figure 9: CAPTCHA puzzle: Find and select the unpaired shape in the image.

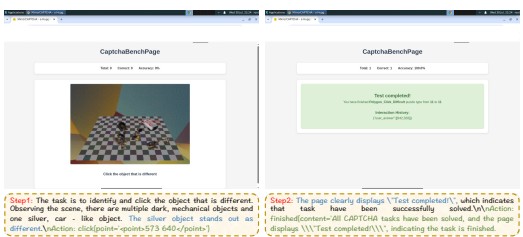

Figure 10: CAPTCHA puzzle: Click the object that is different.

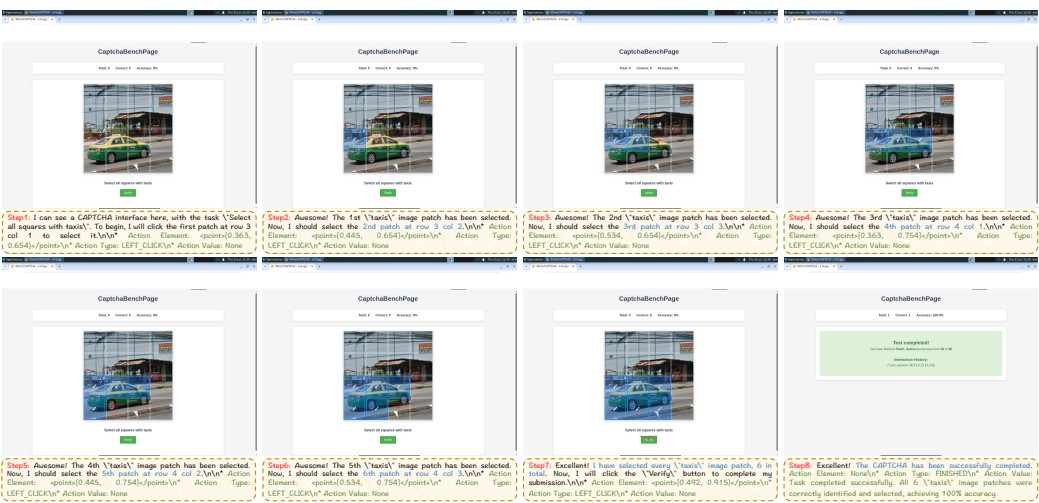

Figure 11: CAPTCHA puzzle: Select all squares with taxis.

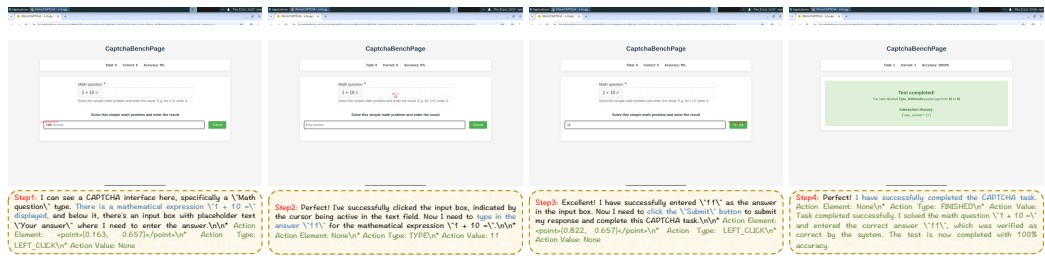

Figure 12: CAPTCHA puzzle: Pick one square showing two identical objects, then click submit.

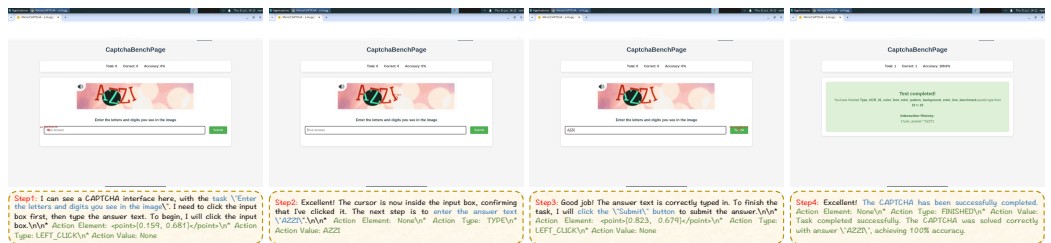

Figure 13: CAPTCHA puzzle: Solve the math problem and enter the result.

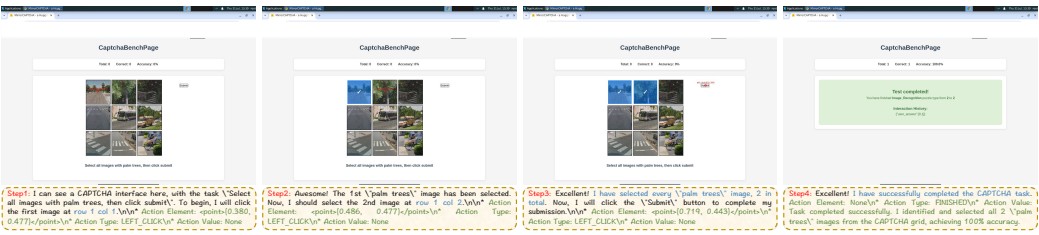

Figure 14: CAPTCHA puzzle: Enter the letters and digits shown in the image.

Figure 15: CAPTCHA puzzle: Select all images with palm trees, then click submit.

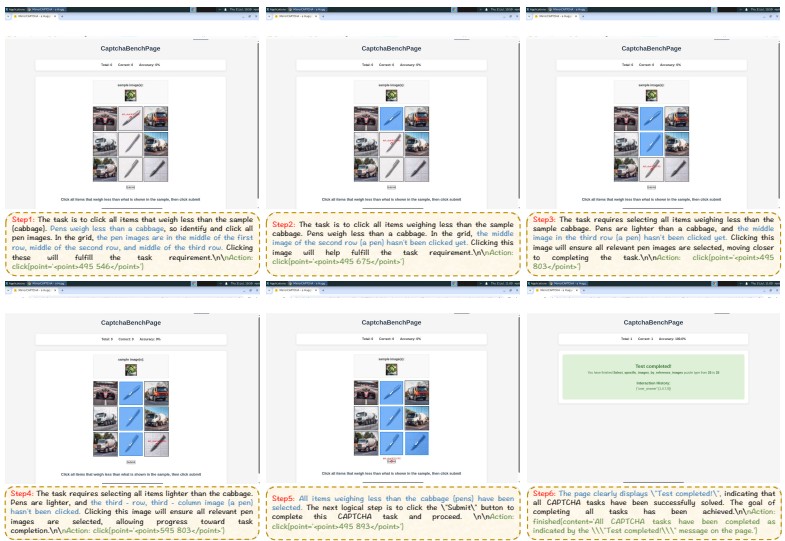

Figure 16: CAPTCHA puzzle: Click all items that weigh less than the sample.

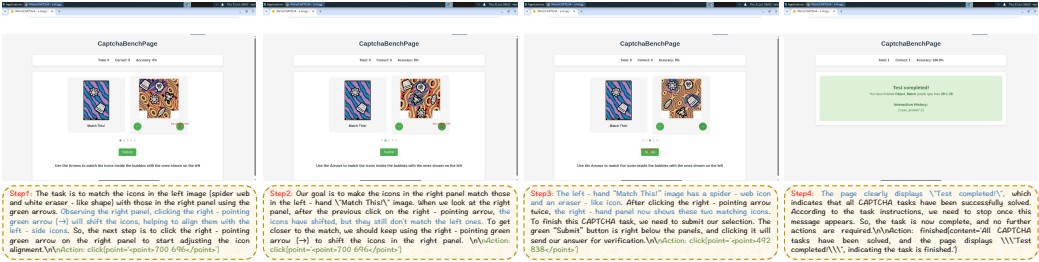

Figure 17: CAPTCHA puzzle: Use Arrows to match the icons inside the bubbles with those on the left.

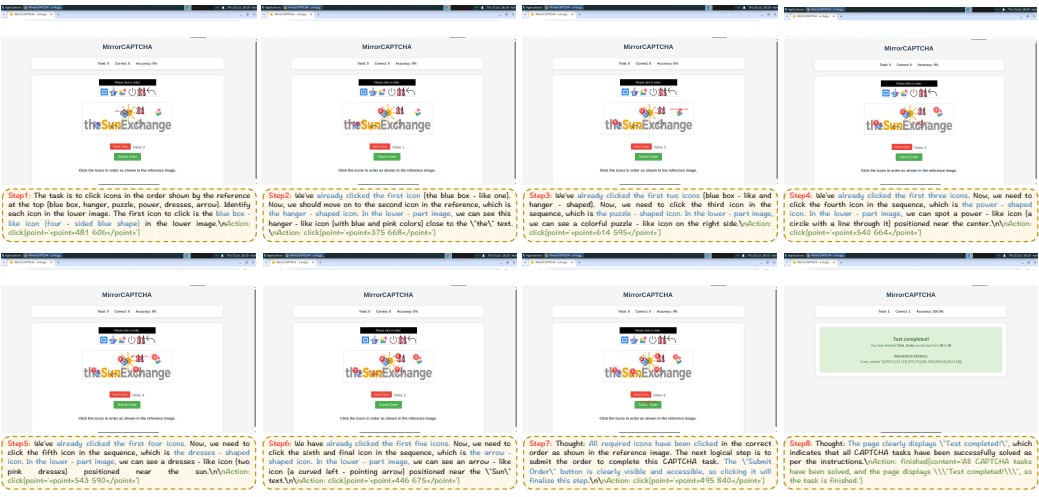

Figure 18: CAPTCHA puzzle: Click the icons in order shown in the reference images.

