# OpenReview forum: "MirrorCAPTCHA: Wild CAPTCHA, Wild Distribution, Wild Web-based Platform Meet Multimodal LLM Agents"
_ICLR.cc/2026/Conference — ICLR 2026 Conference Withdrawn Submission_

### Official Review · Reviewer_XW7P · 2025-10-28

**Soundness:** 2
**Presentation:** 2
**Contribution:** 3
**Rating:** 2
**Confidence:** 5

**Summary:**

This paper introduces MirrorCAPTCHA, a benchmark designed to evaluate multimodal LLM agents on CAPTCHA-solving tasks using real-world distributions. The authors filter 2,095 websites from Common Crawl, classify CAPTCHA into 18 categories, and use random walks to simulate realistic encounter frequencies. They also train Ovis2-Agent-CAPTCHA-8B, which achieves state-of-the-art results, outperforming Gemini-2.5-Pro by 9.4% in Weighted Pass Rate.

**Strengths:**

The model Ovis2-Agent-CAPTCHA-8B is quite interesting.

It is nice to see the focus on failure case analysis.

The use of Common Crawl data and random walks to derive realistic CAPTCHA encounter frequencies is an interesting new idea.

The combination of 18 diverse CAPTCHA types, 1,000 samples, and a web-based evaluation platform provides thorough coverage of real-world challenges.

The introduction of Completion Degree alongside Weighted Pass Rate provides a more complete assessment, capturing partial progress rather than just binary outcomes.

Ovis2-Agent-CAPTCHA-8B demonstrates improvements over open and closed-source models.

The work addresses a genuine bottleneck in web automation that has implications for accessibility, testing, and agent deployment.

**Weaknesses:**

The paper entirely omits discussion of ethical implications and potential for misuse. CAPTCHAs exist specifically to prevent automated abuse of web services. This work essentially provides:
A trained model that breaks security mechanisms
A blueprint for creating CAPTCHA-breaking systems
Data collection via automated triggering of real websites' CAPTCHA.


The related work is short and must be extended. It would be beneficial to have explicit sections on CAPTCHA types and CAPTCHA solving. For the former, given the topic of the paper, a proper explanation of what CAPTCHA is and why they are used is needed. Also, you are using an acronym but never defining it.
For this, you should start with these papers:
https://link.springer.com/chapter/10.1007/3-540-39200-9_18
https://dl.acm.org/doi/abs/10.1145/966389.966390

For example, include these for the latter.
https://ieeexplore.ieee.org/abstract/document/10633630 (https://arxiv.org/pdf/2409.08831)
https://media.kasperskycontenthub.com/wp-content/uploads/sites/63/2017/11/21031220/asia-16-Sivakorn-Im-Not-a-Human-Breaking-the-Google-reCAPTCHA-wp.pdf
https://proceedings.neurips.cc/paper_files/paper/2008/hash/12092a75caa75e4644fd2869f0b6c45a-Abstract.html

Section 3.3 should use proper math to determine the stationary distribution, as this is well defined in probability theory. And if the authors show that they must use the iterative method, then they should show that 10^8 is sufficient and necessary. See https://en.wikipedia.org/wiki/Markov_chain#Finite_state_space and https://stephens999.github.io/fiveMinuteStats/stationary_distribution.html. For the paper, you should cite a probability book or a similar citable source that explains the theory. You can also, in addition, use PageRank https://en.wikipedia.org/wiki/PageRank. This can help make this part shorter.

Your definition of and method to determine cluster types are flawed. Websites often load CAPTCHA from a given provider (e.g., hCaptcha), and the provider then has various kinds of CAPTCHA available, meaning that subsequent visits might get other types. As certain websites have much higher visit probabilities than others, these can greatly impact the weights given to the clusters.

Plesner et al. 2024 [1] show that reCAPTCHAv2 can be fully solved (this corresponds to your type 1 and 2), while Bursztein et al. 2014 [2], and Noury and Rezaei 2020 [3] showed that text-based CAPTCHA can be solved. (The list of examples is not exhaustive.)
It would be nice to include baselines that are not MLLMs. Many agents perform best when they have tools they can call (https://www.anthropic.com/engineering/writing-tools-for-agents and https://modelcontextprotocol.io/docs/getting-started/intro). How would the models perform if given tools that could solve the frequent challenges they encounter?
Assuming the solving rates from [1-3] remain above 98% and the methods handle clusters 1-4 and 6 (there might be more relevant clusters), then the WPR would be (0.35871+0.34847+0.09175+0.03592+0.02814)×0.98=0.8457302. Thus, such a method would outperform your model by 35 percentage points.

[1] https://arxiv.org/pdf/2409.08831
[2] https://www.usenix.org/conference/woot14/workshop-program/presentation/bursztein
[3] https://arxiv.org/abs/2006.08296

There are no measurements of the time to solve a CAPTCHA.

Figures 1 and 2 are hard to read. Please make the font and figures larger.
Also, the captions are generally short and would benefit from being expanded. The tables and figures should be understandable with almost only the caption as context.
Subfigures in Figures 6 and 7 should be cropped only to show what is relevant, but also make the relevant parts larger.

Grammar:
“We then use a modified version WebVoyager” should be “We then use a modified version of WebVoyager” on line 151.
Commas need to be inside the quotation marks. This is relevant for lines 381, 382, 412, etc.

Typos:
Figure 3 caption: MirrirCAPTCHA -> MirrorCAPTCHA

Minor things:
Please sort the examples in Figure 2 so it is easier to find a specific one.
Reference Figure 2 in Table 1.
In Tables 1 and 2, please include a column with the CAPTCHA cluster number (i.e., row number).
For Table 2, make it clear that the average is the weighted average according to the visit probabilities.
You can remove “Categories are ordered by traffic: 1. Image Recognition, 2. Patch Select, 3. OCR Grey, 4. OCR Grid, 5. Drag Drop, 6. OCR ColorPure, 7. Image Match, 8. Rotation Match, 9. Slide Puzzle, 10. OCR ColorPattern, 11. Semantic Select, 12. Semantic Area, 13. OCR Dot, 14. Reference Match, 15. Click Order, 16. Slide Line, 17. Type Arithmetic, 18. OCR Gradient.” from the caption of Table 1.



The core idea of using real-world distributions is valuable, and the benchmark construction is ambitious. However, critical methodological flaws (clustering approach, missing baselines, inadequate probability theory) and presentation issues significantly weaken the contribution.

The current version requires major revisions; however, given IRB approval and other revisions, I will likely increase my score.

**Questions:**

“The resulting corpus comprises 10,000 valid websites spanning diverse domains, including entertainment, media, and social network platforms.” How do you get the domain of a website?

“Standard user actions (e.g., direct registration or login by users) often do not trigger CAPTCHA, as such actions are typically not flagged as suspicious.” You need to justify this statement, either with results or references. I have anecdotally found this not to be the case, and it contradicts what you write in the caption of Figure 3.

You define the types of CAPTCHA you find. However, what are the providers? According to this website https://trends.builtwith.com/widgets, some of the leading providers are Google reCAPTCHA, Cloudflare, and hCaptcha.

How do you handle reCAPTCHAv3 and similar ones? These do not trigger unless the user behaviour is abnormal; however, they are silent in the background and do not pose a challenge to the user. reCAPTCHAv3 uses reCAPTCHAv2 as a fallback for challenges.
If agent A performs poorly on reCAPTCHAv2 but very well on v3, it might still be better than agent B, which performs well on v2 but always fails/triggers v3.

Why did you go for a non-uniform sample distribution? (cf. Table 1).

For the completion degree, how do you get the ground truth labels? E.g. for images this requires hand labelling (as stated by the Breaking reCAPTCHAv2 authors in a later work: https://arxiv.org/pdf/2409.05558)

Why is Table 3 missing several rows?

How stable is the CAPTCHA distribution over time?

**Details Of Ethics Concerns:**

This paper raises several interconnected ethical concerns: (1) Security/Safety. They develop methods to break CAPTCHA security mechanisms without discussing risks or safeguards; (2) Harmful methodologies. They provide trained models and blueprints for CAPTCHA-breaking systems; (3) Legal compliance. Automated interaction with 2,095 live websites likely violates terms of service; (4) Responsible research - no IRB approval mentioned.

---

### Official Review · Reviewer_LFqG · 2025-10-30

**Soundness:** 3
**Presentation:** 3
**Contribution:** 3
**Rating:** 6
**Confidence:** 3

**Summary:**

This paper looks at the problem that autonomous web agents may encounter when tasks they wish to perform are blocked by CAPTCHAs. In such cases, the agent myst be able to solve the CAPTCHA in order to proceed and complete its task. To address this problem, the authors first develop MirrorCAPTCHA, a benchmark dataset of CAPTCHAs designed to be a “mirror” of real-world CAPTCHA distribution that agents might encounter. Second, the paper proposes a metric for agent success on CAPTCHAs and presents a synthetic pipeline to train models on representatives of the kinds of CAPTCHAs they would encounter in web tasks. Measurements in the paper suggest that the authors model trained using this pipeline outperforms previous models.

**Strengths:**

This is a realistic practical problem. The paper makes good progress and will be of interest to those developing web agents that may encounter CAPTCHAs. The dataset and the probability distribution that is developed in section 3.3 seem reasonable and well thought out.

**Weaknesses:**

It is not clear that the evaluation metrics correspond exactly to meaningful success of a web agent. For example, it seems that partially solving a CAPTCHA that guards access to a web page will not be enough to gain access. If that is the case, then why is partial solution helpful? Better experimental evaluation would be useful.

**Questions:**

Have you considered web sites that allow visually imparied users, for example, to select an audio CAPTCHA in place of a visual one? Some sites do so for accessibility. In this case, it is not necessary to solve the harder visual CAPTCHA.

---

### Official Review · Reviewer_47VV · 2025-11-03

**Soundness:** 2
**Presentation:** 2
**Contribution:** 3
**Rating:** 2
**Confidence:** 3

**Summary:**

This paper introduces MirrorCAPTCHA, a benchmark that collected 2,095 websites with active CAPTCHAs from Common Crawl, and categorizes them into 18 types using K-means. They propose two evaluation metrics: Weighted Pass Rate and Completion Degree.
Also, they develop Ovis2-Agent-CAPTCHA-8B, which achieves SOTA performance on the benchmark.

**Strengths:**

This paper focuses on an important area that has not been discussed often in previous benchmarks. Releasing this dataset would help future research in this area.

**Weaknesses:**

- The paper title is vague and confusing.

- Clustering CLIP embeddings of screenshots, then refining and splitting until reaching 18 buckets, is weakly justified. There are no metrics for cluster quality or stability. Since evaluation depends on interaction semantics rather than image appearance, K-means seems to be unnecessary.

- The paper introduces a new model trained on their own dataset, then reports SOTA on that benchmark, which might be biased toward the dataset. Including evaluation on Open CaptchaWorld and MCA-Bench would further verify the results.

**Questions:**

Some suggestions:
- Add a "Limitations and Ethical Considerations" section.
- Please add more information on hyperparameters and reproducibility details.

**Details Of Ethics Concerns:**

Actively mining and replicating live CAPTCHA raises anti-abuse, TOS, and dual-use concerns that the paper only discusses briefly.

---

### Note · Authors · 2026-01-04

I have read and agree with the venue's withdrawal policy on behalf of myself and my co-authors.